# Multichannel Feedforward Active Noise Control System with Optimal Reference Microphone Selector Based on Time Difference of Arrival

**Kenta Iwai** [1,*] **, Satoru Hase** [2] **and Yoshinobu Kajikawa** [2]

[1] College of Information Science and Engineering, Ritsumeikan University, Shiga 525-8577, Japan
[2] Faculty of Engineering Science, Kansai University, Osaka 564-8680, Japan; k222388@kansai-u.ac.jp (S.H.); kaji@kansai-u.ac.jp (Y.K.)
* Correspondence: iwai18sp@fc.ritsumei.ac.jp or kaji@kansai-u.ac.jp

**Abstract:** In this paper, we propose a multichannel active noise control (ANC) system with an optimal reference microphone selector based on the time difference of arrival (TDOA). A multichannel feedforward ANC system using upstream reference signals can reduce various noises such as broadband noise by arranging reference microphones close to noise sources. However, the noise reduction performance of an ANC system degrades when the noise environment changes, such as the arrival direction. This is because some reference microphones do not satisfy the causality constraint that the unwanted noise propagates to the control point faster than the anti-noise used to cancel the unwanted noise. To solve this problem, we propose a multichannel ANC system with an optimal reference microphone selector. This selector chooses the reference microphones that satisfy the causality constraint based on the TDOA. Some experimental results demonstrate that the proposed system can choose the optimal reference microphones and effectively reduce unwanted acoustic noise.

**Keywords:** multichannel ANC system; feedforward control; time difference of arrival; causality constraint

## 1. Introduction

Acoustic noise has become a serious problem with the increased use of industrial equipment, such as engines, manufacturing plants, air conditioners, and so forth. One of the methods used to reduce unwanted noise is passive noise control, such as ear plugs, which cannot reduce noise at low frequencies. Another approach is the use of active noise control (ANC) systems [1–4]. ANC systems use an adaptive digital filter [5–7] to track the time-varying characteristics of the noise source and acoustic environment. The control structure of an ANC system is classified into feedforward [2,8,9] control and feedback control [10–12].

A feedforward ANC system consists of a reference microphone, an error microphone, and a loudspeaker called the secondary source. The reference microphone and error microphone are used to pick up the unwanted noise and to sense the residual noise, respectively. Then, anti-noise is generated by the noise control filter and radiated from the secondary source. Feedforward ANC can reduce unwanted acoustic noise such as broadband noise by placing the reference microphone close to the noise source [4].

In general, an ANC system uses the filtered-x algorithm [13,14], which requires a secondary path model. The secondary path model is an identified secondary path between the secondary source and the error microphone. Hence, if the secondary path changes, then the performance of the ANC system deteriorates. To solve this problem, head-mounted ANC systems were proposed, in which a

compact loudspeaker as the secondary source and the error microphone are located near the user's ears [15–21].

If multiple noise sources are distributed in a wide area, the noise reduction performance of the single-channel feedforward ANC system deteriorates. Therefore, a multichannel feedforward ANC system [22–29] is necessary to reduce multiple unwanted noises. For the multichannel ANC system, the locations of the reference microphones are very important because the causality constraint [30,31] should be satisfied to maintain the noise reduction performance. The causality constraint is that the unwanted acoustic noise propagates to the control point faster than the anti-noise. If the causality constraint is violated, then the noise reduction performance of the feedforward ANC system deteriorates. This problem occurs when some reference microphones are not sufficiently close to the noise source. In particular, the noise reduction performance deteriorates if the arrival direction of the unwanted noise changes owing to the movement of the noise source.

To solve this problem, we propose a multichannel ANC system with an optimal reference microphone selector [32]. The proposed system estimates the time difference of arrival (TDOA) between the unwanted acoustic noise at the control point and the unwanted acoustic noise at the reference microphones located around the control point. On the basis of the estimated TDOA, appropriate reference microphones that satisfy the causality constraint are selected. Then, the multichannel feedforward ANC system is operated using the selected reference microphones.

This paper is organized as follows. In Section 2, the principle of the single-channel feedforward ANC system and its causality constraint are explained. Then, the principles of the multichannel ANC system and the proposed ANC system with the optimal reference microphone selector based on the TDOA are described in Section 3. Experimental results are given in Section 4 and a conclusion is given in Section 5.

## 2. Single-Channel Feedforward ANC System and Causality Constraint

### 2.1. Single-Channel Feedforward ANC System

A principle of the ANC system is based on the superposition between the unwanted noise and the anti-noise, which is generated by the noise controller. The configuration of the ANC system depends on the noise properties. A feedforward ANC system can reduce broadband noise and consists of two microphones, one loudspeaker, and one noise controller. The microphone near the noise source is called the reference microphone and is used to pick up the unwanted noise. The other microphone near the noise control region is called the error microphone and senses the residual noise $e(n)$, expressed as:

$$
\begin{aligned}
e(n) &= d(n) - y'(n) \\
&= d(n) - s(n) * y(n) \\
&= d(n) - s(n) * \left[ \mathbf{w}^{\mathrm{T}}(n)\mathbf{x}(n) \right],
\end{aligned}
\tag{1}
$$

where $d(n)$ is the unwanted noise through the primary path (In general, the primary path is defined as the path between the reference microphone and the error microphone. In this paper, we define the primary path as the path between the noise source and the error microphone for the discussion in Section 3.) $P$, $s(n)$ is the impulse response of the secondary path $S$ at sample time $n$, $\mathbf{w}(n) = [w_0(n)\ w_1(n) \cdots w_{N-1}(n)]$ is the filter coefficient vector of the noise control filter $W$, $\mathbf{x}(n) = [x(n)\ x(n-1) \cdots x(N-1)]$ is the reference signal vector obtained by the reference microphone, and $N$ is the filter length of the noise control filter.

The noise controller calculates the filter coefficients of the noise control filter. Many ANC systems use the filtered-x algorithm [13,14] to update the filter coefficients. A block diagram of the feedforward ANC system is shown in Figure 1. In general, the anti-noise propagates either the error microphone and the reference microphone. The acoustic path from the secondary loudspeaker to the reference microphone is called a feedback path. The feedback path cannot be ignored when the ANC system

is adopted to the duct or the secondary loudspeaker generates large anti-noise. Fortunately, in some ANC applications like a head-mounted ANC system, the feedback path can be ignored because the secondary loudspeaker is located near the user's ears and does not require generating large anti-noise. In this paper, the filter coefficients of the noise control filter are updated by the filtered-x normalized least mean square (FXNLMS) algorithm, whose update equation is given by:

$$\mathbf{w}(n+1) = \mathbf{w}(n) + \frac{\alpha}{\beta + \|\mathbf{u}(n)\|^2}\mathbf{u}(n)e(n), \tag{2}$$

where $\alpha$ is the step size parameter, $\beta$ is the regularization parameter, $\mathbf{u}(n) = [u(n)\ u(n-1)\cdots u(N-1)]$ is the filtered reference signal vector obtained as:

$$u(n) = \hat{\mathbf{s}}^{\mathrm{T}}\mathbf{x}(n), \tag{3}$$

$\hat{\mathbf{s}} = [\hat{s}_0\ \hat{s}_1\ \cdots\ \hat{s}_{L-1}]$ is the impulse response of the secondary path model $\hat{S}$, and $\|\cdot\|$ denotes the norm.

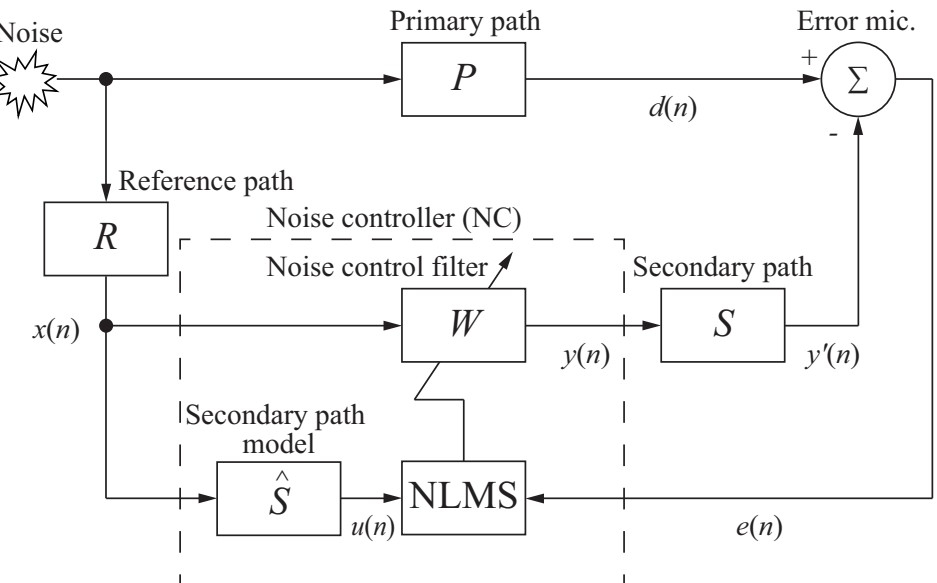

**Figure 1.** Block diagram of a single-channel feedforward active noise control (ANC) system.

## 2.2. Causality Constraint of Feedforward ANC System

In the feedforward ANC system, the anti-noise is generated by the convolution of the noise control filter and the reference signal obtained by the reference microphone. Then, the anti-noise is radiated from the secondary source and cancels the unwanted noise. The anti-noise should reach the error microphone earlier than the unwanted noise. In other words, the total delay of the reference path, the noise controller, and the secondary path should be smaller than that of the primary path. This is called the causality constraint, and if this condition is not satisfied, the causality constraint is violated and the feedforward ANC system cannot reduce the unwanted noise. This causality constraint can be represented by:

$$D_{\mathrm{P}} > D_{\mathrm{R}} + D_{\mathrm{C}} + D_{\mathrm{S}}, \tag{4}$$

where $D_{\mathrm{P}}$ and $D_{\mathrm{R}}$ are the propagation delay from the noise source to the error microphone and reference microphone, respectively. $D_{\mathrm{C}}$ is the processing delay due to the calculations to generate the anti-noise. $D_{\mathrm{S}}$ is the response delay of the analogue-to-digital converter, digital-to-analogue converter,

analogue low-pass filters, loudspeaker used for the secondary source, and the propagation delay between the secondary loudspeaker and the error microphone. Here, inequality (4) can be rewritten as:

$$D_\mathrm{P} - D_\mathrm{R} > D_\mathrm{C} + D_\mathrm{S}. \tag{5}$$

The left-hand side of inequality (5) represents TDOA between the reference microphone and the error microphone.

## 3. Multichannel Feedforward ANC System with Optimal Reference Microphones Based on TDOA

### 3.1. Multichannel Feedforward ANC System

If the multiple noise sources are located in a wide area, it is difficult to reduce the unwanted noises using the single-channel feedforward ANC system. A multichannel feedforward ANC system is used to reduce the unwanted noises in this situation. In general, a multichannel feedforward ANC system has multiple reference microphones, secondary sources, and error microphones. Figures 2 and 3 respectively show the structure and a block diagram of a multichannel feedforward ANC system. In Figures 2 and 3, $J, K$, and $M$ represent the numbers of reference microphones, secondary sources, and error microphones, respectively. Here, this composition is called a case $(J, K, M)$ ANC system. Although there are feedback paths from the secondary sources to the reference microphones in a real situation, the feedback paths are neglected to simplify the discussion. In Figure 3, $\mathbf{P}, \mathbf{R}, \mathbf{W}$, and $\mathbf{S}$ depict the primary path matrix, reference path matrix, coefficient matrix of the noise control filters, and secondary path matrix, respectively. Here, the multichannel feedforward ANC system uses the multiple reference FXNLMS (MRFXNLMS) algorithm to update the filter coefficient vector $\mathbf{w}_{k,j}(n)$, and the update is given by:

$$\mathbf{w}_{k,j}(n+1) = \mathbf{w}_{k,j}(n) + \frac{\alpha \sum_{m=1}^{M} \mathbf{u}_{m,k,j}(n)e_m(n)}{\beta + \sum_{k=1}^{K} \sum_{j=1}^{J} \left\| \mathbf{u}_{m,k,j}(n) \right\|^2}, \tag{6}$$

where

$$\mathbf{w}_{k,j}(n) = [w_{k,j,0}(n)\, w_{k,j,1}(n) \cdots w_{k,j,N-1}(n)]^\mathrm{T}, \tag{7}$$

$$\mathbf{u}_{m,k,j}(n) = \left[ u_{m,k,j}(n)\, u_{m,k,j}(n-1) \cdots u_{m,k,j}(n-N+1) \right]^\mathrm{T}, \tag{8}$$

$$u_{m,k,j}(n) = \hat{\mathbf{s}}_{m,k}^\mathrm{T} \mathbf{x}_j(n), \tag{9}$$

$$\mathbf{x}_j(n) = \left[ x_j(n)\, x_j(n-1) \cdots x_j(n-N-1) \right]^\mathrm{T}, \tag{10}$$

$$\hat{\mathbf{s}}_{m,k} = [\hat{s}_{m,k,0}\, \hat{s}_{m,k,1} \cdots \hat{s}_{m,k,L-1}]^\mathrm{T}, \tag{11}$$

$$e_m(n) = d_m(n) - y'_m(n), \tag{12}$$

$$y'_m(n) = \sum_{k=1}^{K} \mathbf{s}_{m,k}^\mathrm{T} \mathbf{y}_k(n), \tag{13}$$

$$\mathbf{s}_{m,k}(n) = [s_{m,k,0}(n)\, s_{m,k,1}(n) \cdots s_{m,k,L-1}(n)]^\mathrm{T}, \tag{14}$$

$$\mathbf{y}_k(n) = [y_k(n)\, y_k(n-1) \cdots y_k(n-L-1)]^\mathrm{T}, \tag{15}$$

$$y_k(n) = \sum_{j=1}^{J} \mathbf{w}_{k,j}^\mathrm{T}(n)\mathbf{x}_j(n). \tag{16}$$

Hereafter, we consider the case $(J, 1, 1)$ system with $J$ noise controllers. In general, the direction of the unwanted noise is unknown. One of the solutions to this problem is for the reference microphones to be located around the noise control point. An example of this situation is noise environment that

occurred in the medical room. In the medical room, a lot of machines are located and they generate the noise. In this case, the timing for generating the noise from these machines are different from each other, i.e., the direction of the unwanted noise changes from time to time. Then, the reference microphone may not satisfy the causality constraint. Even if the multichannel ANC system is used, some reference microphones may not satisfy the causality constraint. Then, the noise reduction performance of the ANC system degrades.

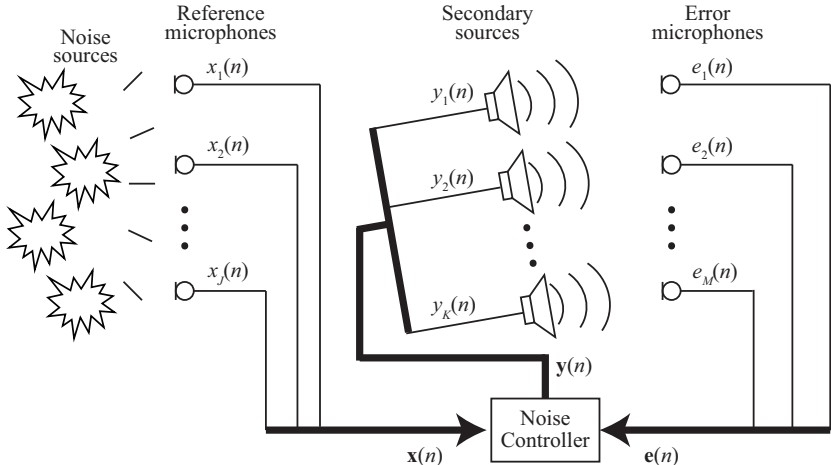

**Figure 2.** Structure of multichannel feedforward ANC system with *J* reference microphones, *K* secondary sources, and *M* error microphones.

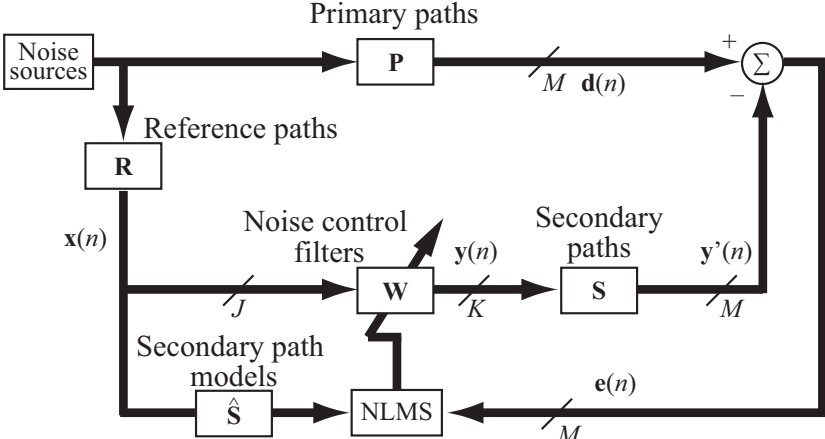

**Figure 3.** Block diagram of multichannel ANC system with *J* reference microphones, *K* secondary sources, and *M* error microphones.

### 3.2. Proposed Multichannel Feedforward ANC System

To solve the above problem, a multichannel feedforward ANC system with an optimal reference microphone selector based on the TDOA is proposed. The proposed ANC system has *J* reference microphones and only a few reference microphones that satisfy the causality constraint are selected. A block diagram of the proposed system is shown in Figure 4. In Figure 4, $\mathbf{v}(n) = [v_{1,1}(n) \cdots v_{m,j}(n) \cdots v_{M,J}(n)]$ represents the switch vector for each reference microphone. In the proposed system, the TDOA is obtained by calculating the cross-correlation between the reference signal obtained at the reference microphone $x_j(n)$ and the unwanted noise obtained at the error microphone $d_m(n)$ as follows:

$$\text{TDOA}\left(x_j(n), d_m(n)\right) = \arg \max_{\tau} \left\{ \sum_n x_j(n+\tau) d_m(n) \right\}. \tag{17}$$

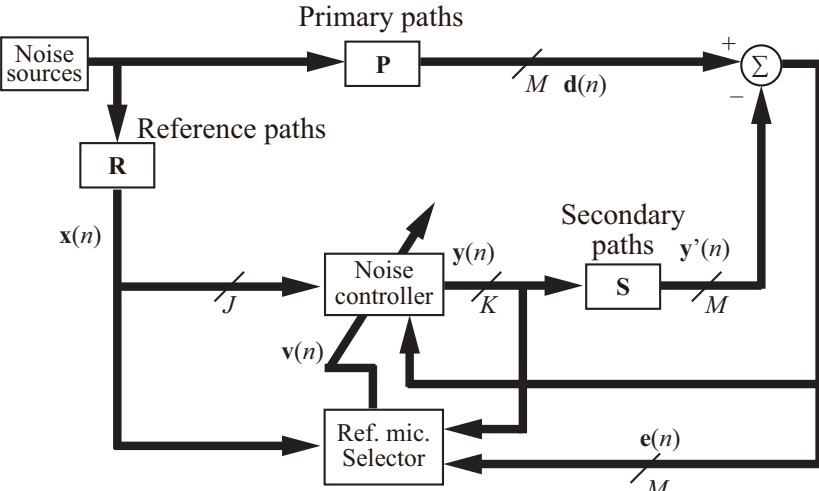

(**a**) Proposed multichannel ANC system with reference microphone selector

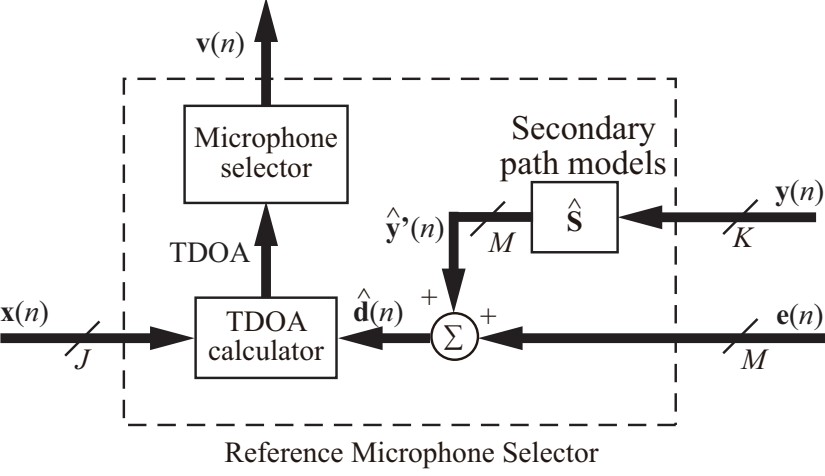

(**b**) Reference microphone selector

**Figure 4.** Block diagram of proposed multichannel feedforward ANC system (case $(J, K, M)$) with reference microphone selector based on TDOA (time difference of arrival).

Then, the causality constraint (5) is rewritten as:

$$\text{TDOA}\left(x_j(n), d_m(n)\right) > D_C + D_S. \tag{18}$$

Here, the time delay of the secondary path $D_S$ cannot be obtained directly. Fortunately, the secondary path model $\hat{S}$ is identified before operating the ANC system and $D_S$ can be estimated from the group delay of $\hat{S}$ as:

$$D_S \approx \min \left\{ -\frac{d\theta_{\hat{S}}(\omega)}{d\omega} \right\}, \tag{19}$$

where $\omega$ is the angular frequency. $D_C$ is the delay due to the calculation of the filter coefficients of the noise control filter. In other words, $D_C$ depends on the performance of the digital signal processor (DSP). The switch vector $\mathbf{v}(n)$ is obtained as follows:

$$v_{m,j}(n) = \begin{cases} 1, \left( \mathrm{TDOA}\left( x_j(n), d_m(n) \right) > D_\mathrm{C} + D_\mathrm{S} \right), \\ 0, \left( \mathrm{otherwise} \right). \end{cases} \tag{20}$$

When $v_{m,j}(n) = 1$, the $j$th reference microphone is selected for the $m$th error microphone, and when $v_{m,j}(n) = 0$, the $j$th reference microphone is not selected for the $m$th error microphone. In the proposed method, the multiple reference microphones are selected using Equation (20). In other words, all the reference microphones that satisfy the causality constraint are selected. Here, the noise control filter is updated as follows:

$$\mathbf{w}_{k,j}(n+1) = \mathbf{w}_{k,j}(n) + \frac{\alpha \sum_{m=1}^{M} \mathbf{u}_{m,k,j}(n) e_m(n) v_{m,j}(n)}{\beta + \sum_{k=1}^{K} \sum_{j=1}^{J} \left\| \mathbf{u}_{m,k,j}(n) \right\|^2}. \tag{21}$$

If the noise environment is changed and the noise reduction performance deteriorates, the reference microphones are reselected. To reselect the reference microphones, the proposed ANC system must detect the change in the noise environment. This is achieved by using the reduction defined by:

$$\mathrm{Reduction} = 10 \log_{10} \frac{\sum_{m=1}^{M} P_{d,m}(n)}{\sum_{m=1}^{M} P_{e,m}(n)}, \tag{22}$$

where

$$P_{d,m}(n) = \sum_{l=0}^{L_\mathrm{P}-1} d_m^2(n-l), \tag{23}$$

$$P_{e,m}(n) = \sum_{l=0}^{L_\mathrm{P}-1} e_m^2(n-l). \tag{24}$$

The reduction is calculated at each sample time. When the reduction becomes smaller than a threshold, it is judged that the noise environment has changed and the reference microphones are reselected.

However, the calculation of the reduction and TDOA is difficult because the unwanted noise $d_m(n)$ cannot be directly obtained. Fortunately, the error signal $e_m(n)$ can be obtained by the error microphone and the output signal of the secondary source $y_m(n)$ can be estimated using the anti-noise $y_k(n)$ and the secondary path model $\hat{\mathbf{s}}_{m,k}$. Hence, $d_m(n)$ is estimated as

$$\hat{d}_m(n) = e_m(n) + \hat{y}'_m(n), \tag{25}$$

$$\hat{y}'_m(n) = \sum_{k=1}^{K} \hat{\mathbf{s}}_{m,k}^\mathrm{T} \mathbf{y}_k(n). \tag{26}$$

Moreover, $P_{d,m}(n)$ and $P_{e,m}(n)$ are calculated from the $L$-sample average of the power as

$$\hat{P}_\theta(n) = \frac{L-1}{L} \hat{P}_\theta(n-1) + \theta^2(n) \tag{27}$$

to reduce the computational complexity.

If the noise environment changes, then the update of the noise control filter is stopped and the propagation of the anti-noise is also stopped to prevent the divergence of the controller. This is because

the optimal noise control filter is changed after the noise environment changes. In general, the optimal noise control filter of the single-channel feedforward ANC system is expressed as

$$W_{\mathrm{opt}}(z) = \frac{P(z)}{R(z)S(z)},$$  (28)

where $W_{\mathrm{opt}}(z)$, $P(z)$, $R(z)$, and $S(z)$ are the optimal noise control filter, primary path, reference path, and secondary path in the $z$-transform representation, respectively. If the noise environment changes, then the primary path and reference path also change. Therefore, the optimal filters before and after changing the noise environment are different. When the update of the noise control filter is stopped, the unwanted noise can be directly obtained by the error microphone. After the optimal reference microphones are selected, the noise control filter is initialized with zeros and updated. A flow chart of the reference microphone selector is shown in Figure 5.

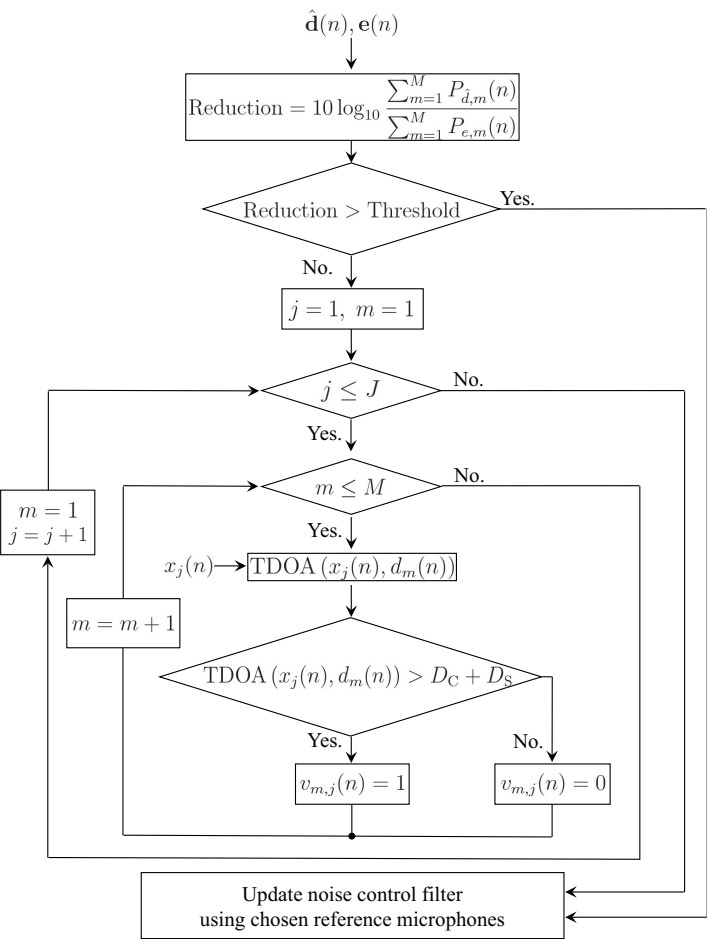

**Figure 5.** Flow chart of the reference microphone selector.

## 4. Experimental Results

We conducted an experiment in a soundproof room (width: 3.1 m, depth: 2.9 m, height: 2.1 m) to evaluate the effectiveness of the proposed method.

In this experiment, we assumed a similar noise environment to that inside a factory or an incubator in a hospital, in which the noise sources have various locations and the unwanted noises are generated with different timings. In a factory, there are various industrial machines around the workers. We focus on a situation in which the workers sit in front of a production line and the ANC system reduces the unwanted noise around the workers' ears. On the other hand, a baby inside an incubator does not move significantly or quickly. There are many medical devices around the incubator that generate

unwanted noise with different timings. In this scenario, the user's ears as the noise control point do not move significantly or quickly. Hence, the movement of the head does not affect the TDOA in this experiment. Moreover, in these environments, the direction of the noise does not frequently change. Therefore, we focused on only gradual changes in the noise environment.

The average sound pressure level of the unwanted noise was about 80 dB at the error microphone. We implemented the ANC system as shown in Figure 6. In this experiment, we used a head-mounted ANC system [18] that measured the primary and secondary paths for the multichannel ANC arrangement shown in Figure 7. The pictures of the measurement arrangements are shown in Figure 8. The secondary loudspeaker used for the head-mounted ANC system is the headphone (PFR-V1, Sony, Tokyo, Japan). Unwanted noises are emitted from the loudspeakers (Eclipse TD-510, Fujitsu-ten, Hyogo, Japan). The reference microphones and error microphone are the electret condenser microphones (AT9904, Audio-Technica, Machida, Tokyo, Japan). Analogue low pass filters are implemented on MSPAMP800 (Hiratsuka Enginieering, Kanagawa, Japan) shown in Figure 6. In Figure 7, the error microphone on the head-mounted ANC does not move significantly or quickly as a result of the head movement. In other words, the head movement does not affect the TDOA in this arrangement. In this arrangement, the differences in the distance between the primary path and the reference path are shown in Table 1. In Table 1, a positive value indicates that the primary path is longer than the reference path (the reference microphone is closer to the noise source than the error microphone) and the causality constraint is satisfied. On the other hand, a negative value indicates the opposite case (the error microphone is nearer the noise source than the reference microphone) and the causality constraint is not satisfied. The secondary path model was identified by NLMS under the conditions shown in Table 2. Here, Figure 9a shows the identified impulse response of the secondary loudspeaker, Figure 9b,c represent the amplitude-frequency response and phase-frequency response of the secondary loudspeaker. Figure 9d shows the group delay which represents the time delay at each frequency. From Figure 9d, the response delay $D_S$ is confirmed to know that the causality constraint is satisfied or not in the experimental arrangement. In the following experiments, the step size parameter and regularization parameter were determined via preliminary experiments to obtain the highest noise reduction performance under each condition.

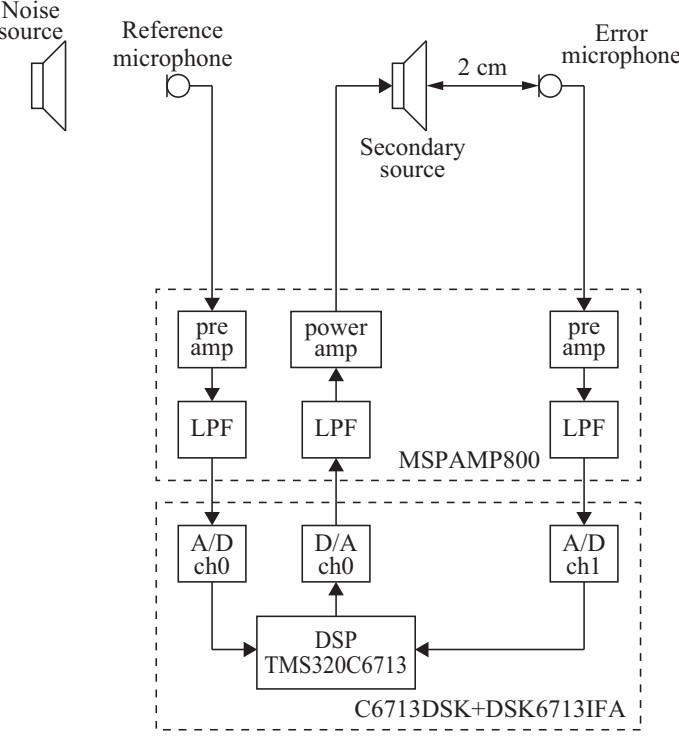

**Figure 6.** Implementation of the system.

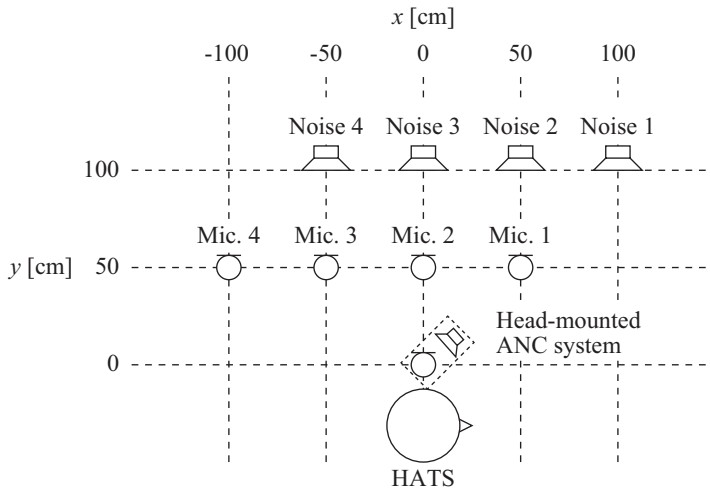

**Figure 7.** Arrangement of equipment in the experiment.

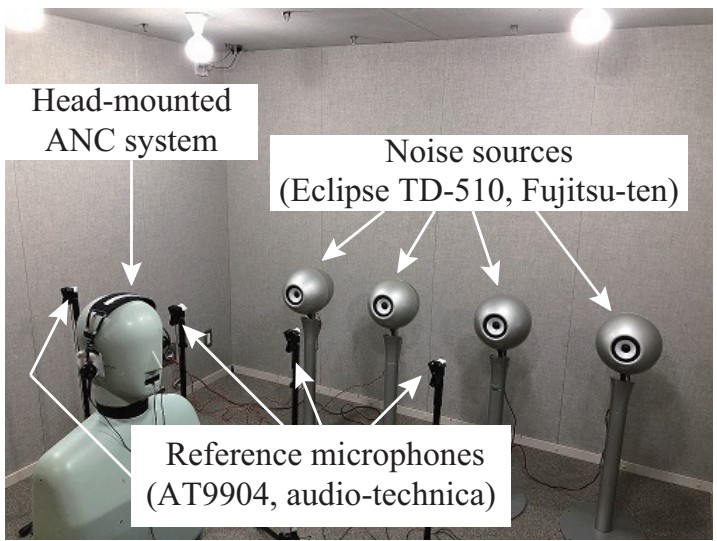

(**a**) Experimental environment

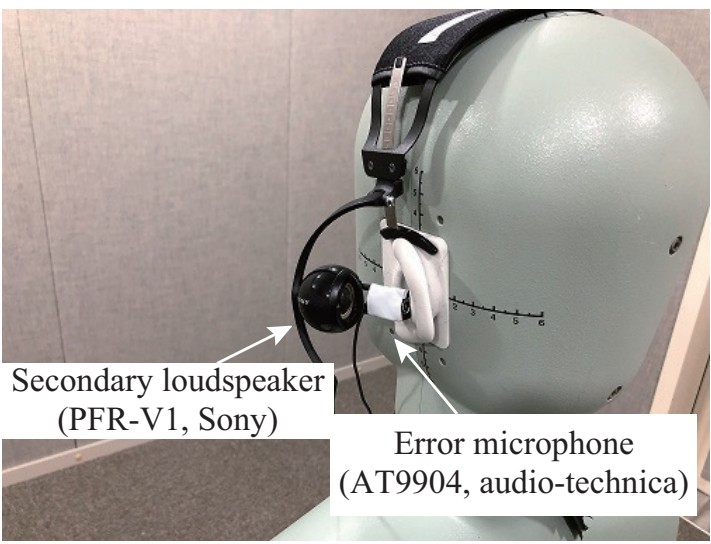

(**b**) Head-mounted ANC system

**Figure 8.** Pictures of the experimental arrangement.

**Table 1.** Differences in distance between primary path and reference path [cm].

|                   | Noise 1 | Noise 2 | Noise 3 | Noise 4 |
|-------------------|---------|---------|---------|---------|
| Reference mic. 1  | 70.7    | 61.8    | 29.3    | 0.0     |
| Reference mic. 2  | 29.6    | 41.1    | 50.0    | 41.1    |
| Reference mic. 3  | −16.7   | 0.0     | 29.3    | 61.8    |
| Reference mic. 4  | −64.7   | −46.3   | −11.8   | 41.1    |

**Table 2.** Identification conditions of the secondary path.

| | |
|---|---|
| Input signal | White noise |
| Sampling frequency | 8000 Hz |
| Cutoff frequency of analogue low-pass filter (LPF) | 2000 Hz |
| Tap length of secondary path model $L$ | 100 |

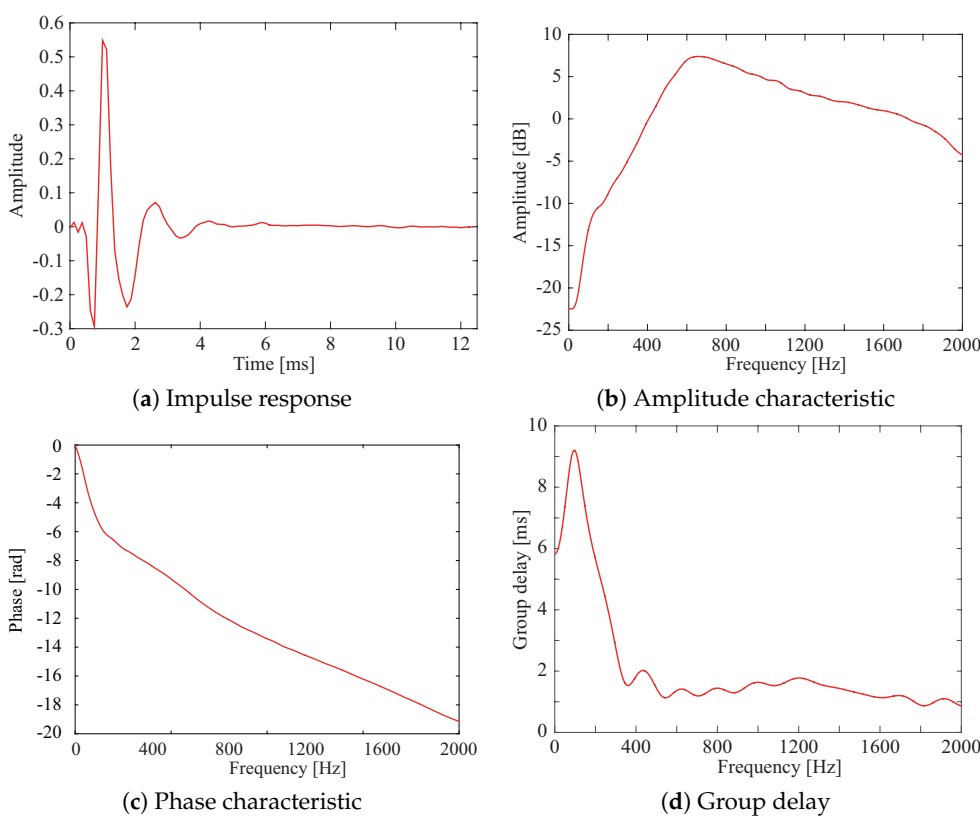

(**a**) Impulse response

(**b**) Amplitude characteristic

(**c**) Phase characteristic

(**d**) Group delay

**Figure 9.** Impulse response and frequency response of the secondary path.

### 4.1. Experiment 1: Noise Reduction Experiment to Evaluate the Multichannel Feedforward ANC System

We first conducted a noise reduction experiment under the conditions shown in Table 3 to evaluate the multichannel feedforward ANC system with multiple reference microphones. The step size parameter is set to 0.01 to converge the noise control filter because the background noise exists in the real environment. In this experiment, we used one to four reference microphones and the ANC system was started after 5 s. The average reduction (22) per second was used as the evaluation score. The experimental results are shown in Figures 10–12. Figure 10 shows the noise reduction performance of the single-channel ANC system. From Figure 10, the ANC system with a single reference microphone can achieve noise reduction when the primary path is longer than the reference path. Moreover, the noise reduction ability is greatest when the reference microphone used is the one closest to the noise. In this arrangement, reference microphone 1 is the closest to noise sources 1 and 2, reference microphone 2 is the closest to noise source 3, and reference microphone 3 is the closest to noise source 4.

Figure 11 shows the noise reduction performance of the case (2, 1, 1) multichannel ANC system with two reference microphones: the most effective reference microphone in Figure 10 and another one. From Figure 11, the multichannel ANC system can reduce the unwanted noise more effectively the single-channel ANC system. Moreover, the multichannel ANC system with the nearest and next-nearest reference microphones shows the best noise reduction performance and highest convergence speed.

Figure 12 shows the noise reduction performance of the case (3, 1, 1) and (4, 1, 1) multichannel ANC systems with three or four reference microphones. The case (3, 1, 1) system uses the two most effective reference microphones in Figure 11 and one other reference microphone. From Figure 12, the noise reduction ability of the case (3, 1, 1) multichannel ANC system is almost the same as that of the case (2, 1, 1) system. This is because some reference microphones do not satisfy the causality constraint, which affects the noise reduction performance and convergence speed. Therefore, the reference microphones that satisfy the causality constraint should be selected to obtain better noise reduction performance of the multichannel ANC system.

**Table 3.** Experimental conditions of noise reduction experiment 1.

| Input signal | White noise |
|---|---|
| Sampling frequency | 8000 Hz |
| Cutoff frequency of LPF | 2000 Hz |
| Tap length of noise control filter $N$ | 400 |
| Tap length of secondary path model $L$ | 100 |
| Step size parameter $\alpha$ | 0.01 |
| Regularization parameter $\beta$ | $1.0 \times 10^{-6}$ |

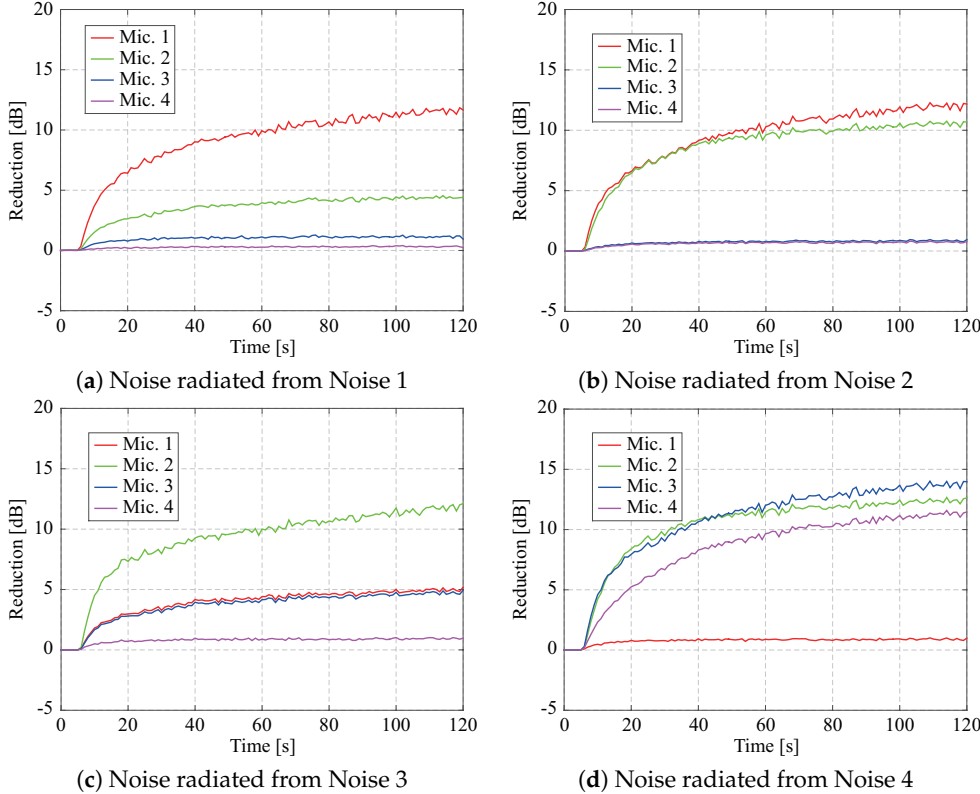

(a) Noise radiated from Noise 1

(b) Noise radiated from Noise 2

(c) Noise radiated from Noise 3

(d) Noise radiated from Noise 4

**Figure 10.** Noise reduction performance of a single-channel feedforward ANC system in the experiment.

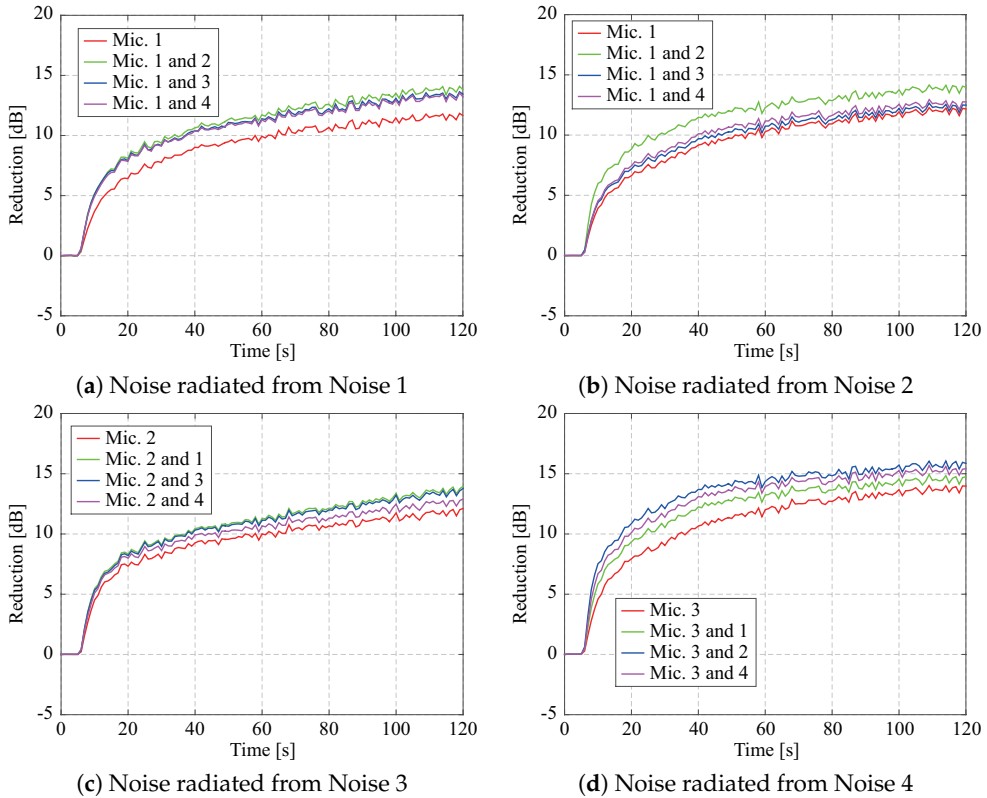

**Figure 11.** Noise reduction performance of a two-channel feedforward ANC system in the experiment.

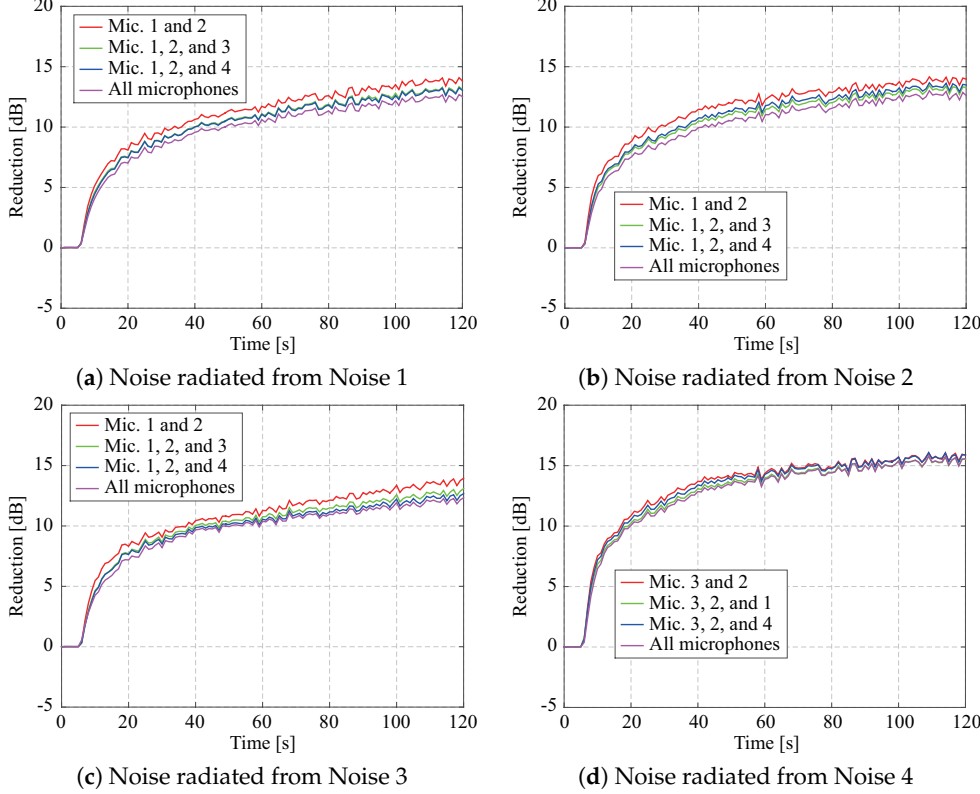

**Figure 12.** Noise reduction performance of three- or four-channel feedforward ANC system in experiment.

*4.2. Experiment 2: Detection of Change in Noise Environment and Selection of Reference Microphones in the Proposed ANC System*

Next, we conducted a numerical experiment to evaluate the noise reduction ability and the selection of the optimal reference microphones of the proposed ANC system in the case that the noise environment changes. The experimental arrangement is the same as that in Figure 7 and the simulation conditions are shown in Table 4. The secondary path model is also the same as that shown in Figure 9. From Figure 9d, the minimum group delay is 0.86 ms, which is used as the delay of the secondary path $D_S$. The delay due to the calculation to generate the anti-noise $D_C$ is zero because this experiment was conducted numerically.

**Table 4.** Simulation conditions for experiment 2.

| | |
|---|---|
| Noise | White noise |
| Sampling frequency | 8000 Hz |
| Cutoff frequency of LPF | 2000 Hz |
| Tap length of noise control filter $N$ | 400 |
| Tap length of secondary path model $L$ | 100 |
| Step size parameter $\alpha$ | 0.2 |
| Regularization parameter $\beta$ | $1.0 \times 10^{-6}$ |

The noise was generated from noise source 1 from 0 to 100 s, from noise source 2 from 100 to 200 s, from noise source 3 from 200 to 300 s, and from noise source 4 from 300 to 400 s. The reference microphones were reselected when the reduction given by Equation (22) was smaller than 1 dB; in other words, the threshold for the reduction was set to 1 dB. This is because a reduction of less than 1 dB means that the noise level is higher than that in the case of no control. The simulation results are shown in Figure 13. From Figure 13a, the unwanted noises are reduced by the proposed ANC system. Moreover, from Table 1 and Figure 13b, the reference microphones that satisfy the causality constraint are selected in each region. From all of the experimental results, the proposed multi-channel ANC system is effective for noise reduction under the condition that the direction of the noise changes.

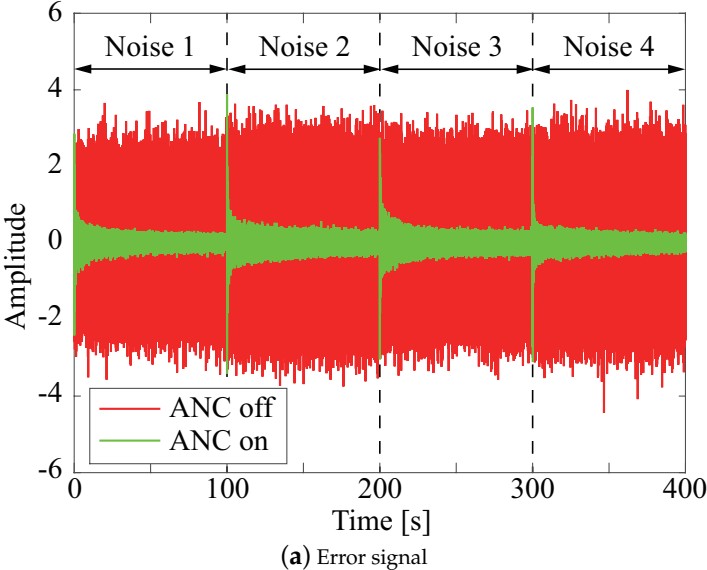

(**a**) Error signal

**Figure 13.** *Cont.*

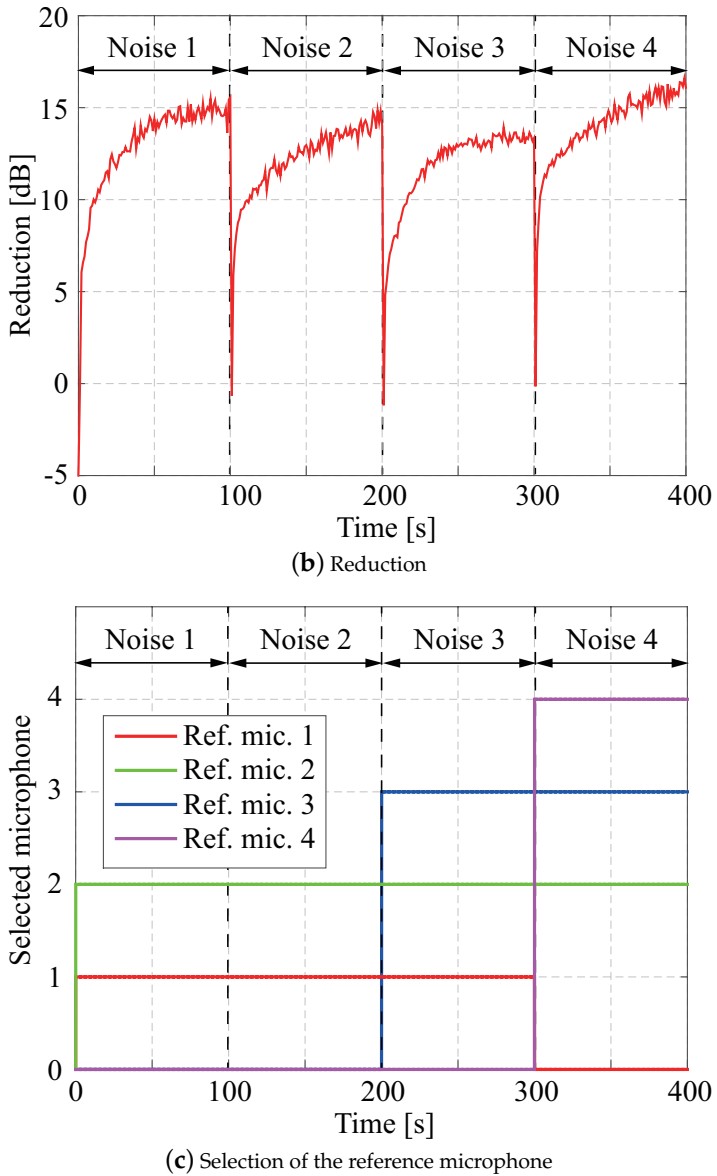

**Figure 13.** Noise reduction performance in cases where arrival direction of noise is changed by using optimal reference microphones.

## 5. Conclusions

In this paper, we proposed a multichannel feedforward ANC system with an optimal reference microphone selector based on the TDOA. The noise reduction performance of a multichannel feedforward ANC system deteriorates when the noise environment such as the noise arrival direction changes. This is because some reference microphones do not satisfy the causality constraint. To solve this problem, we proposed a multichannel feedforward ANC system with an optimal reference microphone selector. This selector chooses the reference microphones that satisfies the causality constraint on the basis of the TDOA. Then, the reference microphones for which the TDOA is larger than the total delay comprising the computational time and the delay of the secondary path are chosen. The detection of the change in the noise environment uses the amount of noise reduction called the reduction. The experimental results demonstrate that the proposed system can choose the reference microphones that satisfy the causality constraint and effectively reduce unwanted noise.

In the future, we will examine the noise reduction performance in the case that multiple unwanted acoustic noises are simultaneously generated from various directions. Moreover, in the

proposed system, the threshold for the reduction directly affects the selection of the reference microphones. Hence, we will examine the effect of the threshold for the reduction in detail.

**Author Contributions:** Y.K. conceived of the proposed method. S.H. developed the method and conducted the experiments. K.I. added the data to show the detail of the experiment, wrote this manuscript, and modified the figures in the manuscript. All authors discussed the results and contributed to the final manuscript.

**Funding:** This work is supported by the MEXT (Ministry of Education, Culture, Sports, Science and Technology)-Supported Program for the Strategic Research Foundation at Private Universities, 2013-2017 (S1311042) and JSPS (Japan Society for the Promotion of Science) KAKENHI (15K00256).

**Conflicts of Interest:** The authors declare no conflict of interest associated with this manuscript.

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
