# Peer review of "Multichannel Feedforward Active Noise Control System with Optimal Reference Microphone SelectorBased on Time Difference of Arrival"

_applsci, doi:10.3390/app8112291_

Round 1
Reviewer 1 Report
The paper is very well written with a clear structure, detailed description so that even no-experts can understand it easily. The idea of selecting proper reference mic is very interesting and have been proven with simple but convincing results. A few extensions of this work has been mentioned in the conclusions, and the review highly encourages the authors to investigate the robustness of the proposed method in a more complex real-life environment.
Some minor comments:
1. In P5, could you clarify why (J,1,1) is considered in this paper? Can you give some examples of such a system with the situation you mentioned, i.e., reference mics are placed near the noise source and the direction of the noise source could change. This would help readers to better appreciate the merit of the proposed system.
2. It would be good if some photos of the implemented system together with the testing environment are included so that the readers can better understand the experiment setup,
Author Response
Thank you for reviewing our paper.
Reply for the reviewer 's comments are attached in this system
and please check it.

Reviewer 2 Report
 When the noise environment has changed, the noise reduction performance of a single-channel ANC system degrades. The authors propose a multichannel feedforward ANC (Active Noise Control) system with an optimal reference microphone selector based on the time difference of arrival. The selector of a multichannel feedforward ANC system chooses the reference microphones that satisfy the causality constraint.The main conclusions of this work show that the proposed system can choose the optimal reference microphones and effectively reduce unwanted acoustic noise.
 As a whole, the reviewed work contains the results that may be of some interest to the scientific community. This paper may be published after the following remarks will be taken into account.
Remarks:
1. line 89-91: Please show the explanation of the variable more clear.
2. line 144-145: Please explain the background of the expression(15) in detail.
3. line 183: Authors should add explanation as a result of Fig.8.
4. line 194-195: What is the reason that the step size parameter is different with Table 3 and Table 4?
Author Response
Thank you for reviewing our paper.
The reply for the reviewer 2's comments are attached in this system
and please check it.

Reviewer 3 Report
The submitted paper presents interesting results related to active noise control systems. An approach of beneficial selection of reference signal sensors based on time difference of arrival is presented. Theoretical introduction to the method has been given. Experiments have been conducted to evaluate the performance of the proposed method. I believe that the paper constitutes a contribution to the field and can be accepted in the current form.
Author Response
Thank you for reviewing our paper. The reviewers 1, 2, and 4 have given us the advises for the minor revision and we have modified our paper.
Reviewer 4 Report
It is a well-organized and clearly written paper with its own purpose.
But it has its own limitation because it covers only (J, 1, 1) case.
Followings are minor comments to improve the paper quality:
1) In experiments, there's no information on microphones, speakers, head-mounted ANC system and LPF filters used. More explanation will help to understand the overall experiments.
2) Page 11, in Fig. 8 (c), unwrapped version of phase characteristic will be more understandable.
Author Response
Thank you for reviewing our paper.
The reply for the reviewer 4's comments are attached in this system
and please check it.
